# Oxidative Potential Versus Biological Effects: A Review on the Relevance of Cell-Free/Abiotic Assays as Predictors of Toxicity from Airborne Particulate Matter

**DOI:** 10.3390/ijms20194772

**Published:** 2019-09-26

**Authors:** Johan Øvrevik

**Affiliations:** 1Department of Air Pollution and Noise, Division of Environmental Medicine, Norwegian Institute of Public Health, 0213 Oslo, Norway; johan.ovrevik@fhi.no; Tel.: +47-21-07-64-08; 2Department of Biosciences, Faculty of Mathematics and Natural Sciences, University of Oslo, 0316 Oslo, Norway

**Keywords:** air pollution, particulate matter, oxidative stress, health effects, inflammation, cellular responses, mechanisms of effects

## Abstract

*Background and Objectives*: The oxidative potential (OP) of particulate matter (PM) in cell-free/abiotic systems have been suggested as a possible measure of their biological reactivity and a relevant exposure metric for ambient air PM in epidemiological studies. The present review examined whether the OP of particles correlate with their biological effects, to determine the relevance of these cell-free assays as predictors of particle toxicity. *Methods*: PubMed, Google Scholar and Web of Science databases were searched to identify relevant studies published up to May 2019. The main inclusion criteria used for the selection of studies were that they should contain (1) multiple PM types or samples, (2) assessment of oxidative potential in cell-free systems and (3) assessment of biological effects in cells, animals or humans. *Results*: In total, 50 independent studies were identified assessing both OP and biological effects of ambient air PM or combustion particles such as diesel exhaust and wood smoke particles: 32 in vitro or in vivo studies exploring effects in cells or animals, and 18 clinical or epidemiological studies exploring effects in humans. Of these, 29 studies assessed the association between OP and biological effects by statistical analysis: 10 studies reported that at least one OP measure was statistically significantly associated with all endpoints examined, 12 studies reported that at least one OP measure was significantly associated with at least one effect outcome, while seven studies reported no significant correlation/association between any OP measures and any biological effects. The overall assessment revealed considerable variability in reported association between individual OP assays and specific outcomes, but evidence of positive association between intracellular ROS, oxidative damage and antioxidant response in vitro, and between OP assessed by the dithiothreitol (DDT) assay and asthma/wheeze in humans. There was little support for consistent association between OP and any other outcome assessed, either due to repeated lack of statistical association, variability in reported findings or limited numbers of available studies. *Conclusions*: Current assays for OP in cell-free/abiotic systems appear to have limited value in predicting PM toxicity. Clarifying the underlying causes may be important for further advancement in the field.

## 1. Introduction

Airborne particulate matter (PM) represents one of the major environmental risk factors for disease and premature death worldwide, and has been associated with development or exacerbation of a number of adverse health effects including asthma, chronic obstructive pulmonary disease (COPD), cancer, cardiovascular disease (CVD), metabolic and neurological disorders and adverse birth effects [1,2,3,4,5,6]. Activation of inflammatory reactions is considered a central driving mechanism for many adverse health effects from particle exposure [3,7,8,9]. In addition, particle exposure may affect cell proliferation, induce cell-cycle alterations, damage DNA and other macromolecules and cause different forms of cell death [10,11,12,13].

The exact mechanisms through which PM trigger these biological effects in exposed cells or organisms, remains to be determined. PM is highly complex and may carry an abundancy of well-known toxins including transition metals, polycyclic aromatic hydrocarbons (PAHs), quinones and bacterial endotoxins, which are considered potential mediators of PM-toxicity [9]. However, a dominating theory is that particles trigger cellular effects through the formation of reactive oxygen species (ROS) or oxidation of biomolecules. The oxidative stress paradigm in particle toxicology encompasses both primary oxidative effects from the particles or adhered particle components, as well as secondary ROS production in exposed cells or tissues. In the strictest form, the oxidative stress paradigm postulates that the biological reactivity of a particle is due to its oxidative potential (OP): The ability to produce ROS or oxidize target substrates directly in contact with biological fluids or cellular molecules. Although non-oxidant mediated mechanisms of PM-toxicity are well known [9], the concept of OP as determinant for PM-toxicity has received considerable attention and it has been linked to all fields of particle toxicology and all the main outcomes of particle exposure, as extensively reviewed by others [14,15,16,17,18,19].

The OP of particles can be measured in cell-free/abiotic systems, and has been suggested as a promising screening method to predict the biological reactivity of particles [20,21]. Acellular OP has also been suggested as a toxicological relevant feature of ambient air PM that could provide improved exposure metrics in epidemiological studies, compared to more conventional particle-metrics such as mass, surface area or number concentrations [22,23,24]. However, the concept of OP as a triggering mechanism for particle toxicity lacks clear, precise definitions. While terms such as oxidative potential or oxidative capacity are most frequently used to describe the particles inherent (primary) abilities to produce ROS or oxidize target substrates directly in contact with biological fluids or cellular molecules, some also incorporate the ability to induce intracellular ROS levels and oxidative stress responses in exposed cells or tissues. There is a significant difference between these two events. While the former represents a potential triggering mechanism of effects, the later often arise from stimulating endogenous production of ROS or RNS (reactive nitrogen species) and may therefore rather be seen as an effect of the exposure [9]. To avoid further confusion, the term “oxidative potential” or “OP” will in this paper strictly be used to describe the inherent abilities of particles or particle-bound components to produce ROS or oxidize substrates, thus measurable under abiotic/cell-free conditions.

To further complicate the scenario, the term ROS also lack precision as it covers a variety of reactive species including hydrogen peroxide (H_2_O_2_), superoxide (O_2_^●−^), singlet oxygen (^1^O_2_), hydroxyl radicals (^●^OH), ozone (O_3_), hypohalous acids (HOX) and organic peroxides. These different species varies considerably in reactivity and longevity, and may elicit different effects in cells and tissues. As pointed out elsewhere [25,26], they are also difficult to distinguish and quantify due to low specificity of available assays (Table 1). Many organic compounds such as PAHs are redox-active only after metabolic activation by cellular enzymes, and are thus not detected by current acellular OP assays. Finally, it should also be considered that establishing a clear causal link between OP in abiotic systems and biological effects is inherently difficult, as antioxidants does not clearly distinguish between the role of primary oxidative events induced by the particles, and the secondary endogenous redox responses of exposed cells [9]. Due to these uncertainties and potential caveats, the precise role of OP in particle toxicity should be interpreted with some caution. The purpose of this paper was therefore to review to what extent OP in cell-free/abiotic systems provides a relevant measure of PM-toxicity, correlating with the biological effects observed in PM-exposed cells, animals and humans.

## 2. Literature Search and Selection of Studies

PubMed was searched for publications containing the terms “particles” or “particulate matter” in combination with the terms “oxidative” or “reactive oxygen” in the title or abstract published up to April 2019. The search “particles[Title/Abstract] AND oxidative[Title/Abstract]” alone, retrieved 3606 publications. The search results were then reduced by adding terms such as “oxidative potential”, “acellular”, “air pollution”, “PM”, “combustion particles”, “diesel exhaust”, “DEP”, “wood smoke”, etc., or common assays for detection of OP (“ESR”, “EPR”, “DTT”, etc.). However, due to inconsistent use of terminology and the massive amount of literature on particles and oxidative stress, discriminating between studies assessing acellular OP and cellular redox responses, systematic search strategies proved insufficient to identify relevant publications. Therefore, the reference lists of central reviews on particle toxicity and oxidative stress were also checked for relevant papers, and the Web of Science and Google Scholar databases was used to search the citation network (cited references and citing articles) of identified studies for other relevant publications.

The main inclusion criteria used for the selection of studies were that they should contain:(1)Multiple particles types or samples;(2)Assessment of OP in cell-free systems;(3)Assessment of biological effects in cells, animals or humans.

Studies that did not meet all three criteria were omitted. Furthermore, if two or more studies were based on identical particle samples and utilized measures of OP obtained from the same analysis, these were not considered as independent, and therefore grouped and counted as one study.

In total 58 studies were identified exploring both the OP and biological effects of PM and/or combustion particles in cells, animals or humans [27,28,29,30,31,32,33,34,35,36,37,38,39,40,41,42,43,44,45,46,47,48,49,50,51,52,53,54,55,56,57,58,59,60,61,62,63,64,65,66,67,68,69,70,71,72,73,74,75,76,77,78,79,80,81,82,83,84]. Among these were three cases of multiple studies based on the same particle samples and measurements of OP: Two studies on in vitro and in vivo effects of a set of wood smoke and PM samples [45,46], and six studies from a series of PM exposures of human volunteers under the RAPTES (Risk of Airborne Particles: A Toxicological–Epidemiological Hybrid Study) project [67,68,69,70,71,72], and three epidemiological studies by Weichenthal and colleagues [80,81,82]. These were grouped and counted as only three independent studies. Notably, three additional studies from one research group assessing association between modeled OP and hospital visits for respiratory and cardiovascular disease could possibly also have been grouped, but were analyzed separately due to some variability in samples used for OP measurements and modeling [66,75,76]. Furthermore, as this review has assessed experimental studies in cells/animals and human studies separately, one study performing both experimental studies in vitro and epidemiological assessment in humans was counted twice [49]. Thus, in total 50 independent studies were identified (Appendix A): 32 in vitro or in vivo studies exploring effects in cells or animals, and 18 clinical or epidemiological studies exploring effects in humans. Furthermore, 31 studies assessed the association between OP and biological effects by statistical analysis. However, two studies obtained statistically significant associations only after assessing different PM samples separately, and were therefore not considered to show an overall statistically significant association [38,53]. The OP assays most commonly used in the identified studies were electron spin resonance (ESR) with 5,5-dimethyl-pyrroline N-oxide (DMPO) as spin trap, the dithiothreitol (DTT) assay, as well as ascorbic aid (AA) and glutathione (GSH) depletion. Other OP assays (Table 1) were used more sporadically and rarely in studies assessing the association between OP and biological effects by statistical analysis (Appendix A).

In cases where the relationship between OP and biological effects were compared by statistical analysis (such as linear regression), the studies were marked with either “statistical significant correlation/association” or “no statistical significant correlation/association” (Appendix A). In cases where no statistical analysis was available, the studies were marked with either “possible association” or “no apparent association” based on a comparison of the rank order of OP and rank order of potency to induce biological effects induced by the different particulates tested in the study (Appendix A). Importantly, the main conclusions of this review have been based on the 29 studies were correlation between OP and biological effects was assessed by statistical analysis [29,32,43,44,49,50,52,55,56,57,58,59,60,63,64,65,66,67,68,69,70,71,72,73,74,75,76,77,78,79,80,81,82,83,84].

Due to the mixed search strategy applied, this work should not be considered a systematic review. However, the material presented and discussed represents, by far, the most comprehensive overview of studies on OP and biological effects presented to date. A detailed overview of all the 58 identified publications, including PM types/samples, OP assay, concentrations used for OP assays and exposure concentrations, biological endpoints assessed and summary of key findings (association/no association), is given in the two tables in Appendix A. All figures presented below are based on the information available in these two tables.

## 3. Results

### Consistency of Associations Between OP and Biological Effects

Statistical significant associations between OP and biological effects were reported in 22 of the 29 independent studies were statistical analysis were applied [29,32,43,44,49,52,57,58,59,60,63,65,66,74,75,76,77,78,79,80,81,82,83,84]: 10 independent studies reported that at least one OP measure was statistically significantly associated with all endpoints examined [43,49,52,57,63,66,78,79,80,81,83], while additional 12 independent studies reported that at least one OP measure was significantly associated with at least one effect outcome [29,32,44,58,59,60,65,74,75,76,77,82,84]. By contrast, seven studies reported no significant correlation/association between any OP measures and any biological effects [50,55,56,64,65,67,68,69,70,71,72,73] (Figure 1, and Appendix A). As some publications were grouped due to shared PM samples and OP analysis, the total number of publications referenced above, exceeds the number of studies considered as independent. Notably, among the positive associations reported, one study reported statistically significant associations between OP and effects only after excluding a selected sample from the analysis [29], one reported very weak associations, which were increased after excluding a subset of particles for which the OP did not correlate with biological effects [43], and one appeared to be based on unrealistically high in vitro concentrations [52]. These three studies were nevertheless included as positive associations in the analysis below.

Of the 22 studies that reported a statistically significant association between at least one OP measure and at least one or more biological effects, were 10 experimental in vitro or in vivo studies in cells or animals [29,32,43,44,49,50,52,55,56,57,58,59,60], and 11 epidemiological or clinical studies in humans [49,63,64,65,66,67,68,69,70,71,72,73,74,75,76,77,78,79,80,81,82,83,84] (Figure 1). Of the seven independent studies reporting no statistical significant association between any OP measure and any biological effects were three independent experimental in vitro or in vivo studies [50,55,56], and four independent epidemiological or clinical studies [64,65,67,68,69,70,71,72,73] (Figure 1). At glance, this may give the impression of a marked overweight of positive associations between OP and biological effects. It seems likely that this at least in part could be the reason for the general assumption often encountered in the literature, that OP may provide a good prediction PM toxicity. However, an alternative way to present this is that 19 of the 29 studies (65%) showed that OP was not statistically significantly associated with at least one or more biological effects examined [29,32,44,50,55,56,58,59,60,64,65,67,68,69,70,71,72,73,74,75,76,77,82,84].

The identified studies utilized a number of different OP measures and biological endpoints, and most investigated effects on multiple biological effects. To further analyze the data, biological endpoints were grouped in various categories. For experimental studies, endpoints were grouped into DNA damage (non-oxidative), ROS/RNS production (measured in cells or animals), oxidative damage (to DNA, proteins and lipids), antioxidant responses, cytotoxicity, inflammatory reactions and “other endpoints” (CYP1A1-expression and phagocytosis). Results obtained by in vitro and in vivo studies were separated. When analyzing reported statistical associations with these specific outcome groups for all OP assays (pooled), it becomes clear that a greater number of reported positive associations with OP were restricted to associations with intracellular ROS formation, oxidative damage and antioxidant responses in vitro (Figure 2). The number of available studies assessing the association between OP and these three individual outcomes was also relatively low (*n* ≤ 3). No study has reported a statistically significant association between OP and non-oxidative DNA damage (strand breaks, adduct formation, etc.), but two studies found that DNA damage was not associated with OP. For the remaining two specific categories, cytotoxicity and inflammation, there seems to be a slight predominance of negative studies, suggesting no statistically significant association with OP (Figure 2). Assessing the reported associations for specific OP assays and these specific outcome groups further strengthened the notion that reproducible statistical significant associations with OP were limited to the redox-related outcomes (Figure 3, and Appendix A). The only exception was the DTT assay, which also appeared to be positively associated with in vitro cytotoxicity, but only two studies were available for this outcome (Figure 3B). In general, few studies (*n* = 1–3) were available for most combinations of specific OP assays and specific effect outcomes, hampering strong conclusions.

Additional 20 experimental in vitro and in vivo studies assessed OP and biological effects without analyzing the association by statistical methods [27,28,30,31,33,34,35,36,37,38,39,40,41,42,45,46,47,48,51,53,54]. These were also evaluated by comparing the rank order of potency for OP and biological effects induced by the particles. If the rank order of OP and ability to induce a biological effect were comparable, the study was denoted as “possible association”. If the rank order clearly differed the study was denoted “no apparent association” (Appendix A). When adding these 20 identified experimental studies where statistically significant associations were not included to the analysis, a marked predominance of studies suggesting no association between OP and all specific outcome groups can be observed, with the exception of in vitro increases in antioxidant levels (Appendix A). This is also the case when separating this material according to the specific OP assays used (Appendix A). However, the relevance of this should be interpreted with care, as such evaluations may be more prone to subjective interpretations. There is another obvious caveat of the approach: If the rank order of potency differs only for one or a few among many PM samples (i.e., outliers), there could still be an overall statistical significant association between OP and effects. On the other hand, if a statistical significant correlation between OP and a biological effect is present, despite marked differences in rank order of OP and effects, one could also argue that OP not sufficiently accounts for the ability to induce the biological effect in question. Nevertheless, due to the possibility of misinterpretation of these data, the main conclusions of this review was based on the studies were statistical analysis was employed by the authors.

Endpoints assessed in epidemiological and clinical studies were also grouped, but in other categories: Biomarkers of inflammation (airways and systemic), biomarkers of oxidative stress (airways and systemic), pulmonary disease (lung function, asthma/wheeze, chronic obstructive pulmonary disease (COPD) and respiratory illness), CVD (vascular disease, ischemic heart disease (IHD) and coronary heart failure (CHF)), mortality (all cause, respiratory, lung cancer and CVD) and other outcomes (pre-term birth and diabetes). When assessing the reported statistical associations between all OP assays (pooled) and specific outcome groups, there seem to be a marked predominance of reported positive associations between OP and asthma/wheeze, and a slight predominance of positive associations between OP and airway inflammation, lung function and CVD (Figure 4). By contrast, there was a marked predominance of negative associations reported for OP and COPD as well as mortality (Figure 4).

Further dividing the epidemiological and clinical studies by specific OP assays revealed some interesting information on the different assays used. ESR with DMPO was only used in three identified studies, and seems at present to provide limited support for any clear association with effects (Figure 5A). The DTT assay, on the other hand, has been used in seven studies and shows a clear association with asthma/wheeze (four positive and no negative studies). However, three out of the four positive associations between OP and asthma/wheeze were reported by the same research group, using modeled OP (DTT) for the overlapping region and time period [66,75,76], and may therefore not be considered as truly independent observations. Positive associations between DTT and other endpoints including airway inflammation, lung function, vascular disease and “other outcomes” (diabetes) has also been reported, but only one study was available for each endpoint, hampering strong conclusions. For IHD and CHF mixed associations have been reported, but also here the number of studies are limited (Figure 5B). By contrast, studies using the AA-depletion assay have almost consistently failed to find any positive associations between this OP assay and any effect outcomes, in epidemiological and clinical studies (Figure 5C). The GSH-depletion assay shows somewhat more mixed associations, and few available studies for each outcome (*n* = 1–2) hampers any conclusions regarding its potential association with effects in humans (Figure 5D).

## 4. Discussion

Assessment of the identified studies revealed a considerable variability in reported association between individual OP assays and specific outcomes. There seem to be a predominance of positive associations reported for OP and redox-related responses in in vitro cell models, including intracellular ROS generation, oxidative damage to macromolecules and antioxidant response. By contrast, a predominance of studies reported that in vitro cytotoxicity, inflammatory reactions and non-oxidative DNA-damage is not associated with the OP of PM. No experimental animal studies exploring the statistical association between OP and effects were identified. Furthermore, epidemiological and clinical studies in humans suggest a positive association between OP measured by the DDT assay and asthma/wheeze in epidemiological and clinical studies. For all other outcomes assessed, the number studies on the association between a specific OP assay and a specific endpoint was either too low to conclude, or conflicting results were reported from different studies (Figure 3 and Figure 5). Notably the AA-depletion assay has almost completely failed to be associated with effects in all studies on humans, suggesting that this particular assay may be of limited relevance for PM toxicity and effects.

The number and variation of biological endpoints investigated for possible association with acellular OP is strikingly high. Indeed, it may seem unlikely, given the complexity of PM composition, the number of distinct PM sources and the variability in biological effects attributed to PM exposure, that all particle induced effects should be caused by one common triggering mechanism: OP. Indeed, several redox-independent triggering mechanisms for PM toxicity have been described, as discussed later. Nevertheless, virtually all possible effects from particle exposure have to some extent been compared to OP. A general impression is therefore that the search for possible associations between OP and PM toxicity has been performed rather randomly, without clear rational for the choice of biological endpoints included in the respective studies. This possibly reflects a lack of consensus regarding the type of effects that most likely could be caused by direct redox-reactions from particles and particle components, and hence would most likely be associated with acellular OP. The present review may bring some clarification to this issue, as the identified studies appear to show a predominance of positive associations between OP and redox-related outcomes in vitro, and asthma/wheeze in humans. The lack of consistent association between OP and other PM-induced effects could be related to study designs and inherent weaknesses in common techniques for OP assessment, to cell-free assays being unable to recreate in vivo conditions, or simply because many effects of particle exposure are triggered through redox-independent mechanisms. Most likely, the lack of consistency arises from a combination of the aforementioned factors.

### 4.1. Considerations Regarding the Methods Used to Measure OP

A detailed discussion of the strengths and weaknesses of the various acellular assays used for OP measurements was beyond the scope of the present paper, and has been extensively covered elsewhere [26,85,86,87,88,89]. A key problem with most OP assays is that they are not precise enough and do not sufficiently discriminate between different reactive species, and different assays detect different species and with varying efficiency (Table 1). Importantly, there is considerable difference in reactivity, longevity and potential to inflict damage to biomolecules between different types of ROS. Helmut Sies, who pioneered the work on oxidative stress, has emphasized that “Simply to talk of ‘exposing cells or organisms to oxidative stress’ should clearly be discouraged. Instead, the exact molecular condition employed to change the redox balance of a given system is what is important” [90]. Unfortunate, such critical information on “the exact molecular condition employed” when exposing cells, animals or humans to PM cannot be provided by the OP assays currently in use. The different assays also use different units for expressing redox properties, as emphasized by others [87]. Thus, it could be argued that OP assays provide neither quantitative nor qualitative measurements. As consequence, OP measurements of the same particle types obtained by different techniques may report considerably different results. This could likely contribute to the lack of correlation with biological effects and explain some of the discrepancies reported from different studies. In this context introduction of standardized reference particles could be a way forward to increase the comparability of OP measurements reported across different studies.

More specific problems are related to some of the assays. The deoxyribose assay has for instance been highly controversial due to some serious caveats, which may lead to misinterpretation of the findings [86]. Assays such as the DTT test cannot differentiate between redox cycling and “simple” one/two electron oxidations. It has also recently been suggested that quinones and transition metals may catalyze DTT oxidation to an extent not relevant for their effects on biological macromolecule, thus confounding the DTT assay [26].

The only method that provides direct quantification of radical species is ESR (or EPR). ESR can detect and quantify persistent radicals including quinones on combustion particles and surface radicals on quartz, while short-lived radicals like O_2_^●−^ and ^●^OH can be measured more indirectly using a spin trap such as DMPO. However, low sensitivity of ESR may be a potential problem [87]. Moreover, ESR with DMPO is often performed in presence of H_2_O_2_ as substrate for ^●^OH formation, amongst others by Fenton-reactive transition metals. It has been argued that this mimics the conditions during phagocytosis, where engulfed dust particles will be in contact with H_2_O_2_ [29]. Whether addition of H_2_O_2_ is relevant for effects on non-phagocytic cells are less clear, as H_2_O_2_ otherwise would not be present in abundancy under normal physiological condition.

Another central question is there whether acellular OP assays can reflect the complexity of in vivo biological conditions. OP tends to be measured under very artificial conditions, often in simple buffers solutions, that do not reflect the composition of the extracellular fluids in the respiratory tract or elsewhere, or the intracellular milieu. Observing that particles or particle components are able to oxidize a single available substrate in a test tube does not necessarily reflect that critical cellular components would be oxidized under real-life conditions when other competing substrates are present, including (but not restricted to) extracellular and intracellular antioxidant. Moreover, the majority of studies reviewed in this paper appear to have performed OP assays at pH 7.4 (if given at all), which is in the pH range of cell growth media and extracellular compartments. Lung surfactant is more acidic and may be in the range of pH 6.6–7.1 [91], and phagocytosed particles encounter a pH of 4.5–5.0 within the lysosomes [92]. It remains unclear whether increased acidity would alter the relative OP of particles. However, the catalytic redox rates of quinones is affected by pH [93,94]. It has also been reported that the DTT assay may be affected by the pH of the assay buffer, producing not only quantitatively, but also qualitatively different outcomes (different rank-order) when assessing the OP of multiple particles in different simulated lung fluids [95]. A number of assay conditions may potentially differ from real-life scenarios, including relevant competition kinetics and sufficient electron donor concentrations to allow proper redox cycling. Thus, there is a need to link what transpires under acellular artificial conditions with the biological and biochemical processes and pathways involved in PM-exposed cells and tissues.

Many studies identified in this review utilized only one concentration of particles for measurements of OP (Appendix A). However, the concentration-dependent formation of ROS by particles in cell-free systems is not necessarily linear [33,96]. This could potentially hamper proper comparisons between particles of high and low OP, and might be problematic in cases when the concentrations used for assessment of OP differed from those used to assess biological effects.

Finally, it should also be considered that the biological assays, which OP has been compared with, could also be flawed, and therefore not truly reflect particle toxicity [26,97]. Issues related to interference between particles and common cytotoxicity test are frequently debated, and carbon-based particulates may also bind cytokines and other proteins hence interfering with Enzyme-Linked Immunosorbent Assay (ELISA) [98,99,100]. The oxidant probe, DCF, may leak back out of cells and not reflect intracellular ROS, and LDH used for assessment of cytotoxicity can be inactivated by oxidation, potentially making the assay prone to errors when used in experiments with high levels of oxidative stress [101].

### 4.2. Redox-Independent Mechanisms of PM Toxicity

One of the most obvious explanations for why OP was not associated with PM-induced effects in several studies is the possibility that OP was not the key driver of the effects in questions. Indeed, a number of redox-independent triggering mechanisms for particle toxicity have been reported, including specific receptor mediated effects and interference with membrane lipids, as reviewed elsewhere [9,102]. The toxicity of ambient air PM is particularly complex and is in part related to soluble constituents, which includes a multitude of reactive organic chemicals, metal ions and biological material, of which many interacts directly with cellular receptors and other signaling proteins [9,102,103]. Zinc, which mediates its effects by binding to cellular enzymes, has already been mentioned. Moreover, proinflammatory effects of coarse PM have in many cases (although not always) been attributed to the presence of bacterial endotoxins and activation of toll-like receptors [34,35,68,104,105]. Zinc and endotoxin content would not affect the outcome of acellular measurements of OP. Furthermore, organic compounds do not only mediate their effects through oxidative stress. PAHs and dioxin-like compounds are well known to activate the cytosolic aryl hydrocarbon receptor (AhR), which in its classical mode of action regulates the expression of Phase 1 metabolizing enzymes, such as CYP1A1, -1A2 and -1B1. CYP1-mediated metabolism may produce a number of redox-reactive metabolites from parent compounds that are not directly measured in acellular OP assays. AhR also possesses several other physiological functions. Amongst others, it seems to play a central role in the regulation of inflammation and immune responses [106,107]. There is an extensive crosstalk between AhR and the proinflammatory transcription factor NF-κB [108], and AhR may also be directly involved in the transcriptional regulation of proinflammatory genes [109,110,111,112,113,114]. AhR and PAHs also appear to play a central role in CVD, as reviewed elsewhere [115,116,117], providing a potential link between PM exposure and adverse effects. These are other possible components and effects not picked-up by acellular assays for OP.

While both oxidant and non-oxidant mediated triggering mechanisms for PM toxicity exists [9], these different mechanisms may likely follow different dose-response relationships, and their relative importance for the initiation of different PM-induced effects remains unclear. There is a need to clarify whether the levels of ROS produced under abiotic conditions are sufficient to produce the biological effects in question, and whether such concentrations are realistic and relevant. Indeed, the oxidative stress paradigm has been extensively debated in recent years and it has been argued that the levels of exogenous oxidants required to distorting the physiological oxidant-antioxidant balance and cause pathology is order of magnitudes higher than what could be reached under real-life conditions in either health or disease [118]. A central task should therefore be to clarify which triggering mechanisms (oxidant or non-oxidant) are the most important for effects at low PM concentrations, realistic for real-life PM exposure scenarios.

### 4.3. Correlation is not Causation

Any scientist should be well aware that correlations and associations do not confirm causation. Even in the cases when OP appears to be associated with effects, other factors may be the underlying cause. A central question is therefore whether OP could be an indicator of something else?

Sampling of ambient PM before and during the closure and after the reopening of a steel mill in Utah Valley has allowed for examination of the contribution of metal-rich particulates to the toxicity PM. Some studies on Utah Valley PM have indicated a possible association between OP (deoxy ribose assay) and inflammatory responses [28,61]. Both metal chelator (deferroxamine) and antioxidant (DMTU and DMSO) treatment blocked acellular ROS generation and production of proinflammatory cytokines in BEAS-2B cells pointing to a possible role of redox active transition metals [28]. However, other studies do not support a clear causal relation between the OP and biological effects of Utah Valley PM [27,31]. Of notice, Utah Valley PM contained high levels of zinc (twice the level of copper and six-fold higher than iron), and the zinc-level was also considerably reduced by the metal-chelator treatment [31]. Studies by Samet and colleagues suggest that zinc could be an important contributor to the proinflammatory effects of metal rich PM [119]. In contrast to metals such as iron and copper, zinc is not redox active but may elicit effects by inhibiting protein phosphatases [120,121]. In fact, studies using metal ions suggest that only zinc and not copper or iron ions were capable of inducing IL-6 and CXL8 responses in the BEAS-2B cells [121,122]. Thus, reductions in proinflammatory effects of Utah Valley PM observed after metal-chelation treatment, could at least in part be due to a reduction in zinc and other redox-inactive constituents, rather than the corresponding reduction in redox active metals and OP. The Utah Valley studies underscores that correlation or association between two observations does not necessarily imply causality, and that other covariates may be more important to the biological effects than OP. Indeed, strong associations have been reported between zinc concentrations in PM and OP measured by different assays, despite the fact that zinc is not detected by OP assays [58].

In striking resemblance of the Utah Valley PM studies, a study on metal-rich PM_2.5_ from the German smelter area Hettstedt and control PM_2.5_ from a non-industrialized area found that the metal-rich Hettstedt-PM_2.5_ induced stronger pulmonary inflammation in healthy volunteers [62]. Although only two particle types do not allow for correlation analysis, the higher proinflammatory effect of metal-rich PM_2.5_ was accompanied by a higher acellular OP (ESR with H_2_O_2_ and DMPO). It was therefore suggested that the higher concentration of redox-active metals in Hettstedt-PM_2.5_ could be responsible for its increased inflammogenicity [62]. However, the most abundant metal in Hettstedt-PM_2.5_ was zinc (six-fold higher than copper and seven-fold higher than iron). It seems possible that the inflammatory effect of the metal-rich Hettstedt-PM_2.5_, at least in part, could have been due to the high zinc content and not necessarily the OP of the particles, as in the case of the Utah Valley PM.

As already discussed, PAHs are another group of PM constituents considered to be among the potential key drivers of PM toxicity [9,115], which are not measured by acellular OP assays. As with zinc, the PAH content of PM may also be strongly associated with OP [58]. A likely explanation for this is that many redox-active organic chemicals such as quinones and oxy-PAHs are produced alongside PAHs during combustion processes, or may be derived directly from PAHs, for instance through photooxidation. In this aspect, it is interesting to note that OP measured by the DTT assay may be statistically significantly correlated with CYP1A1 expression [58,60]. AhR, the key regulator of CYP1A1, is also the major cellular sensor of PAHs and other nonpolar compounds, but reacts poorly to the more polar quinones and oxy-PAHs. Thus, OP measured by the DTT assay is likely associated with at least some outcomes indirectly, due to PM-bound PAHs co-varying with quinones, oxy-PAHs or other redox active organics detected by DTT.

Whether reported associations between OP and effects are truly due to the direct redox properties of PM or more indirectly through correlation between OP and other redox-independent PM properties remains to be determined. Nevertheless, several epidemiological studies have reported that effects in humans were stronger associations OP than PM mass. This suggests that particle composition may be important for the effects, as pointed out by Strak and colleagues [83]. A series of studies from Ontario, Canada, show that OP (GSH- but not AA-depletion) modified the impact of respiratory illness, myocardial infarctions and birth outcomes [79,80,82]. Thus, it would be of particular interest to clarify whether OP or PM components correlating with OP, could explain some of the between-city heterogeneity in risk estimates reported for PM exposure [123].

### 4.4. The Relative Toxicity of Particles Varies across Different Endpoints and Target Cells or Tissues

Another important aspect is that the relative toxicity of particles may vary depending on endpoints and cellular targets. The proinflammmatory potential of different particles does not necessarily correlate with their ability to induce other cellular responses including intracellular ROS formation and cytotoxicity. Even for a specific type of effects (i.e., inflammatory responses) the rank order of potency among different particle types may vary between different cell types and between in vitro cell culture tests and in vivo animal studies. This has been reported in a number of studies on ambient air PM as well as mineral particles and nanomaterials [105,124,125,126,127,128,129,130,131,132,133,134]. These observations not only suggest the existence of a variety of triggering mechanisms for particle toxicity and that different particle characteristics may act as critical determinants for different toxicological effects, but also that triggering mechanisms may vary between different cell types and target organs. Based on this, it seems unlikely that a single determinant such as OP could be a useful predictor of all particle-induced outcomes.

In extension of the above, Donaldson et al. [97] raised another important issue. In a review on the limitation of intracellular oxidative stress as a predictor of particle toxicity they pointed out that the three pathogenic particle types PM_10_, asbestos and quartz have all been reported to induce comparable effects on intracellular ROS formation in vitro, but in real-life exposure scenarios they induce diverse pathologies [97]. Similarly, it could be argued that also OP in cell-free/abiotic systems is incapable of discriminating between classes of particles with different pathological effects.

## 5. Conclusions

The collective evidence suggest that there is limited or no clear evidence of association between individual OP assays and inflammation, non-oxidative DNA-damage and cytotoxicity in experimental studies (in vitro), as well as COPD, CVD, mortality and biomarkers of inflammation and oxidative stress in epidemiological and clinical studies in humans. However, some evidence exists for positive reproducible associations between OP assays and redox responses including intracellular ROS, oxidative damage and antioxidant responses in vitro, and between OP measured by the DTT assay and asthma/wheeze in humans. The limited association with several outcomes could in part be due to the low precision and considerable variation in the ability to detect different reactive species among the OP assays applied, but also due to the inability of such acellular assays to simulate what occurs under biological conditions in real-life. However, it most likely also reflects that other redox-independent triggering mechanisms are important for PM-induced effects. Clarifying these issues will be important to advance the field.

Finally, it should be appropriate to repeat that correlation alone is insufficient to provide causality to a hypothesis, and that a lack of correlation is equally insufficient for its falsification. There may be a number of reasons why OP correlates with a biological effect in one study but not in another. Additional evidence must therefore also be considered. However, the lack of a consistent association with a broad range of biological effects in cells, animals and humans suggests that current assays for OP of particles in cell-free/abiotic systems may have limited value in predicting toxicity of PM and combustion particles.

## Figures and Tables

**Figure 1 ijms-20-04772-f001:**
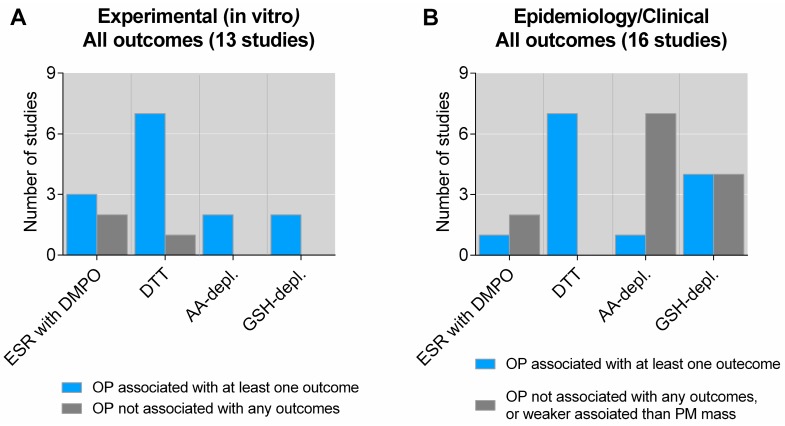
Overview of studies exploring the association between OP and biological effects of particulate matter (PM). The figure displays the number of experimental studies in vitro (**A**) and epidemiological/clinical studies in humans (**B**) exploring the association between specific OP assays and biological effects by statistical analyses. As some studies have explored OP by several different assays, the sum of the individual columns exceeds the total number of studies given in the figure title. Details on the individual studies are given in Appendix A. AA depl.—ascorbic acid depletion; DMPO—5,5-dimethyl-pyrroline N-oxide; DTT—dithiothreitol; ESR—electron spin resonance; GSH depl.—reduced glutathione depletion; OP—oxidative potential.

**Figure 2 ijms-20-04772-f002:**
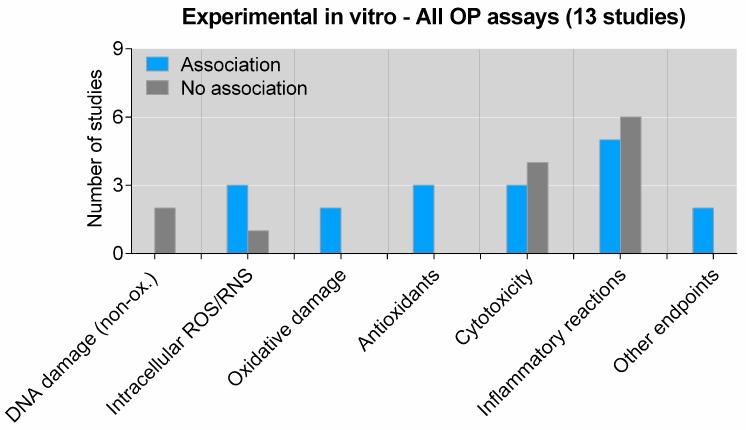
Association between all OP assays (pooled) and biological effects of PM in experimental studies in vitro. The figure displays the number of studies showing an association or no association between OP measured by any assay and specific biological effects in vitro (cell cultures). “Oxidative damage” includes lipid peroxidation and oxidative DNA damage. “Other endpoints” include cellular signaling and proliferation. As some studies have explored association between OP and several different biological effects, the sum of the individual columns exceeds the total number of studies given in the figure title. Details on the individual studies are given in Appendix A. Non-ox—non-oxidative; OP—oxidative potential; ROS—reactive oxygen species; RNS—reactive nitrogen species.

**Figure 3 ijms-20-04772-f003:**
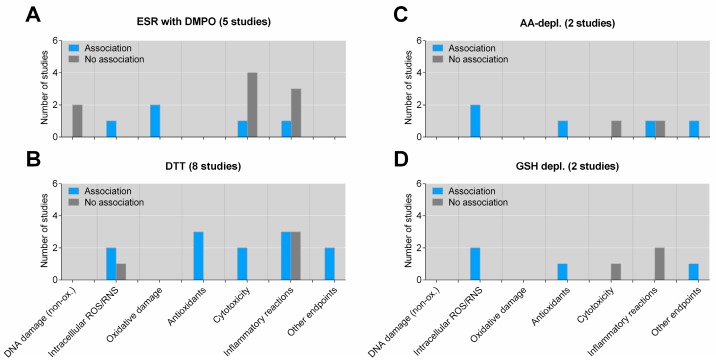
Association between specific OP assays and biological effects of PM in experimental studies in vitro. The figure displays the number of studies showing an association or no association between OP measured by electron spin resonance (ESR) with DMPO as a spin trap (**A**), DTT assay (**B**), ascorbic acid (AA)-depletion (**C**) or reduced glutathione (GSH)-depletion (**D**)and specific biological effects in vitro (cell cultures). “Oxidative damage” includes lipid peroxidation and oxidative DNA damage. “Other endpoints” include cellular signaling and proliferation. As some studies have explored association between OP and several different biological effects, the sum of the individual columns exceeds the total number of studies given in the figure title. Details on the individual studies are given in Appendix A. AA-depl.—ascorbic acid depletion; DMPO—5,5-dimethyl-pyrroline N-oxide; DTT—dithiothreitol; ESR—electron spin resonance; GSH depl.—reduced glutathione depletion.

**Figure 4 ijms-20-04772-f004:**
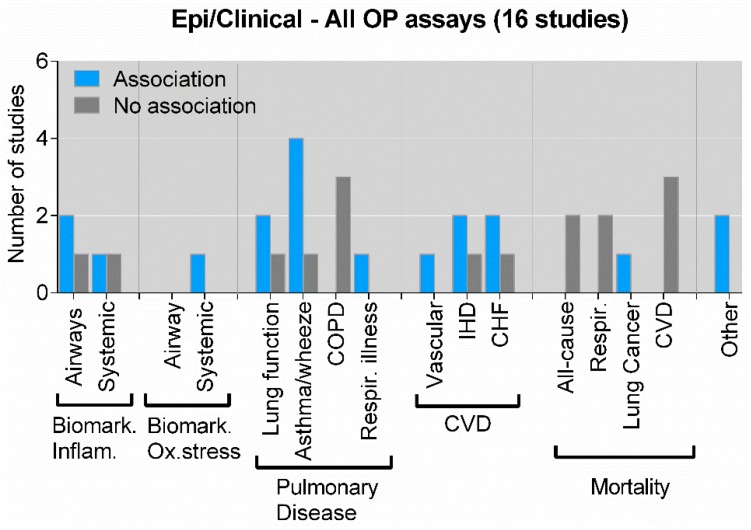
Association between all OP assays (pooled) and biological effects of PM in epidemiological and clinical studies. The figure displays the number of studies showing an association or no association between OP measured by any assay and specific biological effects in humans assessed by epidemiological or clinical studies. “Other endpoints” include pre-term birth and diabetes. As some studies have explored the association between OP and several different biological effects, the sum of the individual columns exceeds the total number of studies given in the figure title. Details on the individual studies are given in Appendix A. Biomark. Inflam.—biomarkers of inflammation; Biomark. Ox.stress—biomarkers of oxidative stress; COPD—chronic obstructive pulmonary disease; CVD—cardiovascular disease; CHF—coronary heart failure; IHD—ischemic heart disease; Respir.—respiratory.

**Figure 5 ijms-20-04772-f005:**
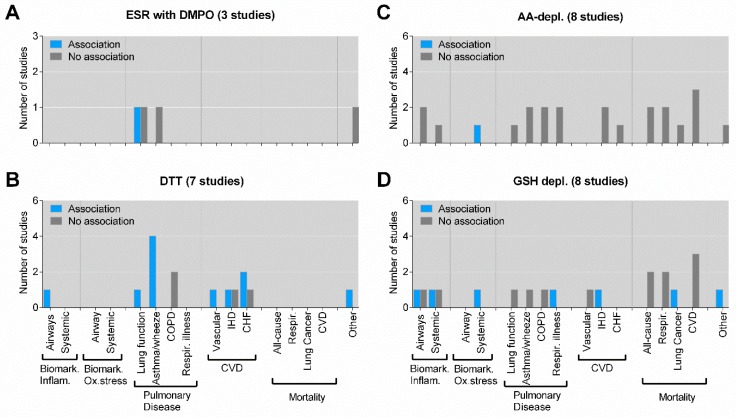
Association between specific OP assays and biological effects of PM in epidemiological and clinical studies. The figure displays the number of studies showing an association or no association between OP measured by ESR with DMPO as spin trap (**A**), DTT assay (**B**), AA-depletion (**C**) or GSH-depletion (**D**), and specific biological effects in humans assessed by epidemiological or clinical studies. “Other endpoints” include pre-term birth and diabetes. As some studies have explored association between OP and several different biological effects, the sum of the individual columns exceeds the total number of studies given in the figure title. Details on the individual studies are given in Appendix A. AA—ascorbic acid; Biomark. Inflam.—biomarkers of inflammation; Biomark. Ox.stress—biomarkers of oxidative stress; COPD—chronic obstructive pulmonary disease; CVD—cardiovascular disease; CHF—coronary heart failure; DMPO—5,5-dimethyl-pyrroline N-oxide; DTT—dithiothreitol; ESR—electron spin resonance; GSH—reduced glutathione; IHD—ischemic heart disease; Respir.—respiratory.

**Table 1 ijms-20-04772-t001:** Assays for detection of the oxidative potential (OP) in cell-free/abiotic systems.

Assay	Species Detected
AA-depletion	Used to measure oxidative potential of transition metals (^●^OH from H_2_O_2_). Interacts with several other reactive species.
GSH-depletion	Most ROS as well as peroxides, alkenals, protein disulfides and sulfenic acids
Congo Red	Hydroxylic, peroxide and hydroperoxide radicals
DCF (DCFH-DA)	^●^NO_2_, ^●^OH, ONOO^−^, peroxyl, aloxyl and carbon-centered radicals, peroxides. Can be used to measure H_2_O_2_ in presence of a peroxidase catalyst (HRP). Prone to photooxidation.
2-deoxyribose	^●^OH and ^●^OH-like species (used as a simple and inexpensive substitute for ESR).
DHE	Can be specific for O_2_^●−^ (require separation of products by HPLC). Interacts with several other reactive species.
DTT	Diverse range of free radicals and reactive species. Reduced by transition metals and quinones, in PM.
ESR (or EPR)	ESR with DMPO as spin-trap measures production of ^●^OH, and is often used in combination with H_2_O_2_. ESR without spin-trapping can be used to measure surface radicals on particles (measures unpaired electrons).
Luminol	HOCl, H_2_O_2_ and ONOO^−^

Abbreviations: AA—ascorbic acid; DCF—dichlorofluorescin; DCFH-DA—dichlorodihydrofluorescein diacetate; DHE—dihydroethidium; DMPO—5,5-dimethyl-pyrroline N-oxide; EPR—electron paramagnetic resonance; ESR—electron spin resonance; DTT—dithiothreitol; GSH—reduced glutathione; H_2_O_2_—hydrogen peroxide; HOCl—hypochlorous acid; HPLC—High-performance liquid chromatography; MDA—malondialdehyde; ^●^NO_2_—nitrogen dioxide; O_2_^●−^—superoxide anion; ^●^OH—hydroxyl radical; ONOO^—^peroxynitrite; PM—particulate matter; ROS—reactive oxygen species.

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
