# Peer review of "Oxidative Potential Versus Biological Effects: A Review on the Relevance of Cell-Free/Abiotic Assays as Predictors of Toxicity from Airborne Particulate Matter"

_ijms, 2019, doi:10.3390/ijms20194772_

Round 1

Reviewer 1 Report

I think that review is a really outstanding contribute to the field of environment pollution. It describes a good correlation betweew airborn Particulate Matter level and healthy problems but, as several studies demonstrated, this correlation could be probably indirect. The strong differences between in vitro (where a clear direct correlation between PM oxidative potential and ROS production is almost defined) and in vivo studies (where there is a high variability in results not showing a significant correlation between PM oxidative potential and biological effects) probably demonstrate an indirect correlation. This original analysis could improve also its novelty by arguing the cause of possible indirect effects of airborn PM oxidative potential on human, plant and animal health. In this case, articles describing studies on the possible direct impact of PM on water, grass, earth ground contamination should be introduced to explain a possible direct correlation between these kind of environmental pollution and food contamination, which directly impact on human health. Otherwise, despite the optional introduction of studies on the direct impact of environmental pollution on food contamination and then, indirectly, on human and animal health, the review could be accepted as it is only after a little check of English language.

Author Response

I thank the reviewer for the positive feedback. Regarding indirect effects of airborne PM oxidative potential on human, plant and animal health, I agree that this is an interesting and important topic. However, I feel that this would be a quite comprehensive task, and beyond the scope of this review, and would rather be suited for a separate paper.

Reviewer 2 Report

The manuscript submitted for review is a literature review of many scientific papers related to the use of oxidative potential as a measure of the biological reactivity of the PM exposure indicator in epidemiological studies.

The manuscript is well written and is suitable for printing in the International Journal of Molecular Sciences, with minor corrections.

General remarks:

Please complete the information on the statistical methods used, list them all, provide their assumptions taken for calculations (e.g. confidence intervals, number of samples - which significantly affects the significance of further inference, etc.). In the chapters ‘Results’ and ‘Discussion’, please cite specific references or authors, on the basis of which individual observations and conclusions were formulated (lines: 140-141, 148, 176, 178, 184, 203, 225, 304, 313, 377-378, 395, 419-420). Some of the keywords used are inappropriate for this manuscript.

Details remarks:

Line 10: words „correspondence” was used twice; Table 1: rewrite the language notation, please; Table 1: DCFH-DA it’s rather should be „dichlorodihydrofluorescein diacetate”, instead „dichlorofluorescin diacetate”.

Regards,

R.

Author Response

I thank the reviewer for the positive feedback. All changes in the MS as shown in yellow in the Highlighted-version of the MS.

General remarks:

Please complete the information on the statistical methods used, list them all, provide their assumptions taken for calculations (e.g. confidence intervals, number of samples - which significantly affects the significance of further inference, etc.).

Response: The number of samples used in the different studies are given in Tables S1 and S2 (supplementary materials). However, in-depth considerations on statistical methods of the individual studies would be a comprehensive task that cannot be done within the given 5-day deadline, and beyond the scope of this review.

In the chapters ‘Results’ and ‘Discussion’, please cite specific references or authors, on the basis of which individual observations and conclusions were formulated (lines: 140-141, 148, 176, 178, 184, 203, 225, 304, 313, 377-378, 395, 419-420).

Response:

I agree with the reviewers comments, and have included references in most places, as suggested. In some cases the sentence in question was rather rephrased, or references to figures or tables were included. In addition, references has also been included in the first paragraph of the Results-section. Specific comments are addressed below.

140-141: This line refers to tables S1 and S2 which should be sufficient

148, 176, 178, 184, 225, and 395: References has been included

203: Reference to Tables S1 and S2 (supplementary materials) has been included

304: Reference to Figs 3 and 5 has been included

313: Sentence has been changed to: “A general impression is therefore that the search for possible associations between OP and PM toxicity has been performed rather randomly…”

377-378: Sentence has been changed to: “Thus, there is a need to link what transpires under acellular artificial conditions with the biological and biochemical processes and pathways involved in particle exposed cells and tissues.”

419-420 : the sentence has been changed to: “There is a need to clarify whether…”

Keywords has been rephrased and Table 1 is updated according to the reviewers suggestion („dichlorofluorescin diacetate” corrected to “dichlorodihydrofluorescein diacetate”)

Sincere regards,

Johan Øvrevik